# STGAN: Detecting Host Threats via Fusion of Spatial-Temporal Features in Host Provenance Graphs

## Abstract

As the complexity and frequency of cyberattacks, such as Advanced Persistent Threats (APTs) and ransomware, continue to escalate, traditional anomaly detection methods have proven inadequate in addressing these sophisticated, multi-faceted threats. Recently, Host Provenance Graphs (HPGs) have played a crucial role in analyzing system-level interactions, detecting anomalous behaviors, and tracing attack chains. However, existing provenance-based detection methods primarily rely on single-dimensional feature analysis, which fails to capture the dynamic and multi-dimensional patterns of modern APT attacks, resulting in insufficient detection performance. To overcome this limitation, we introduce STGAN, a model that integrates spatial-temporal graphs into host provenance graph modeling. STGAN applies temporal and spatial encoding to dynamic provenance graphs to extract temporal, spatial, and semantic features, constructing a comprehensive feature representation. This representation is further fused and enhanced using a multi-head self-attention mechanism, followed by anomaly detection. Through extensive evaluations on three widely-used provenance graph datasets, we demonstrate that our approach consistently outperforms current state-of-the-art techniques in terms of detection performance. Additionally, we contribute to the research community by releasing our datasets and code, facilitating further exploration and validation.

## Keywords

Network Security, Host Provenance Graph, Graph Anomaly Detection

## ACM Reference Format:

Anonymous Author(s). 2018. STGAN: Detecting Host Threats via Fusion of Spatial-Temporal Features in Host Provenance Graphs. In . ACM, New York, NY, USA, 11 pages. https://doi.org/XXXXXXX.XXXXXXX

## 1 Introduction

The complexity and frequency of cyberattacks are rapidly increasing, posing unprecedented threats to businesses, government agencies, and society at large [1]. Modern cyberattacks are no longer confined to malware propagation and data theft but have evolved into highly complex attack chains, encompassing Advanced Persistent Threats (APTs), ransomware attacks, and supply chain attacks. These attacks not only result in substantial economic losses but

also jeopardize the security of critical infrastructure. Traditional anomaly detection methods [22, 32], such as signature-based and rule-based intrusion detection systems, have become inadequate in addressing the challenges posed by these sophisticated attacks.

Host Provenance Graphs (HPGs) are gradually becoming a powerful tool in the field of detecting host threats [6, 13, 27, 47, 49]. By recording and analyzing the interactions between files, network connections, and processes within a host, HPGs construct a comprehensive activity map that helps security analysts identify anomalous behaviors, trace the origins of attacks, and enhance system security. They reveal the attacker's activities within the host and can link multiple attack events, providing critical insights into understanding the attack chain. Recent researchers have proposed various types of provenance-based anomaly detection methods, including static feature analysis, spatial feature analysis, and temporal feature analysis. However, these existing methods still flaw to following limitations:

- **Static Low-Dimensional Feature Analysis:** Early studies [21, 22, 36] extracted statistical features from HPGs or designed heuristic rules for anomaly detection in nodes and edges. However, rules and statistical feature-based mechanisms can easily be mistakenly triggered by benign nodes, so these methods tend to have high false alarm rates and rely heavily on expert knowledge.
- **Temporal Feature Analysis:** Some time-series-based graph methods [18, 29] utilized Graph Sketch to extract statistical feature changes from flow graphs for detecting anomalous nodes or edges. However, these methods focus solely on the statistical information of dynamic graphs, overlooking the rich spatial information within the graph structure.
- **Spatial Feature Analysis:** Some previous research [11, 24, 44] utilized graph embedding methods to extract topological and node features from provenance graphs to identify anomalous nodes and connections. Yet, these methods often overlook the dynamic nature of attacks, making it challenging to capture the temporal evolution of attack behaviors.

Given these limitations, our core idea is to enhance detection performance by integrating temporal and spatial features. This integration poses challenges, as spatial features require whole-graph input for extraction, while temporal features rely on accurate modeling of event sequences. To address these challenges, we propose STGAN, which introduces the concept of spatio-temporal graphs into host anomaly detection for the first time. STGAN combines spatial, temporal, and semantic dimensions to generate comprehensive node representations, enabling more accurate and thorough identification of potential anomalous nodes and malicious behaviors. Specifically, we divide the streaming host audit logs into multiple time steps. Within each time step, we extract edge information from each log entry using regular expressions to form quadruples *<src, dst, type, timestamp>* and construct a provenance graph. For

each time step, we employ Word2Vec [30] and GAT [42] to extract structural and semantic features, respectively, and fuse them into unified spatial features. Temporal features are captured using TGN [39]. To further enhance feature representation, we fuse the spatial and temporal features using a multi-head self-attention mechanism, creating unified feature representations for node anomaly classification. For anomaly detection, we design a lightweight XGBoost detector for downstream anomalous node identification.

In designing STGAN, we specifically addressed the challenges faced by previous methods. (1) STGAN constructs sentences for semantic extraction by using a node's neighbors and edge types, effectively capturing the semantic differences between malicious and benign nodes. This helps us overcome the issue of neglecting semantic features in prior research, ensuring more comprehensive and accurate node embeddings, thereby improving detection performance. (2) We are the first to jointly model and represent both spatial and temporal information of the provenance graph. This enables us to overcome the limitations of earlier research, where single-dimensional feature analysis could not comprehensively capture all relevant characteristics of cyberattacks. (3) The self-attention mechanism further enhances our model by effectively fusing spatial-temporal data, creating a comprehensive feature representation that surpasses previous single-dimensional methods.

We have implemented a prototype of STGAN and conducted extensive evaluations on three three widely-used provenance graph datasets from the DARPA TC E3 [2], including Theia, Trace, and Cadets. Additionally, we selected five *state-of-the-art (SOTA)* methods as benchmarks, including **Threatrace** [44], **Anograph** [9], **Unicorn** [18], **FLASH** [38], and **HOLMES** [32]. The experimental results demonstrate that our method outperforms the current state-of-the-art techniques in terms of performance metrics such as precision and F1 score (exceeding 97%). Furthermore, we conducted ablation studies to validate the rationality and necessity of STGAN's module design. Additionally, we evaluated STGAN's parameter settings to highlight how different configurations affect its performance. In summary, our contributions include:

- **Novel Perspective:** We introduce the concept of spatial-temporal graphs into provenance-based detection for the first time, offering a novel detection perspective for anomaly detection in host provenance graphs.
- **Multi-Dimensional Feature Capture and Fusion:** We leverage spatial and temporal encoders to extract spatio-temporal features from streaming provenance graphs, which are then fused through a multi-head self-attention mechanism to produce more expressive feature representations.
- **Better detection performance:** We built a prototype of STGAN and conducted a comprehensive evaluation on three datasets. The experiments demonstrated that STGAN's detection performance surpasses five different types of *SOTA* methods, including HOLMES [32], Threatrace [44], Unicorn [18], AnoGraph [9], and FLASH [38].
- **Open-Source Resources:** Contributing to the research community by providing open-source datasets and code. [1]

---

[1]STGAN is available at https://anonymous.4open.science/r/STGAN-anonymous.

**Figure 1: The example of host provenance graph.**

**Ethics and Privacy:** All datasets are sourced from public websites, and experiments were conducted in a controlled environment to minimize potential ethical and privacy risks.

## 2 Background

### 2.1 Host Provenance Graph

Host Provenance Graphs have recently been widely used for network threat detection. A host provenance graph represents the activity of processes within a host and is constructed from audit logs (e.g., Windows ETW [5] or Linux Audit [3]). It comprises three types of nodes: process nodes, file nodes, and network nodes, with edges representing system events such as reads, sends, and deletions. Researchers collect audit logs from the target host and extract the basic graph tuples *<src, dst, type, timestamp>* to construct the host provenance graph.

Figure 1 provides an example of a provenance graph where different shape represent different types of nodes, and the edges illustrate the relationships between them. For instance, the relationships between two process nodes may include fork or clone; between process and file nodes, the relationships may include write, read, open, and unlink; and between process and network nodes, the relationships include sendmsg and recvmsg. Overall, the provenance graph records the sequence of activities within the host system, containing rich semantic and spatial information.

### 2.2 Provenance-based IDS

Since the provenance graph can express the relationship between system operating entities in time, existing research has used this feature to build an IDS based on the provenance graph. Including detection schemes based on knowledge labels [20, 22, 32], these schemes construct a series of matching rules based on expert knowledge to match in the origin graph to detect anomalies. Based on the statistics IDS scheme [14, 21, 28], they use the structural feature information of the graph, including: abnormality, discrepancy, time correlation and other features to analyze in the graph to detect anomalies. Recently, more learning-based IDS solutions have been proposed [6, 19, 24, 41, 47, 49]. These solutions use models such as graph representation learning and sequence learning to extract high-dimensional features from graphs to perform anomaly detection in downstream tasks.

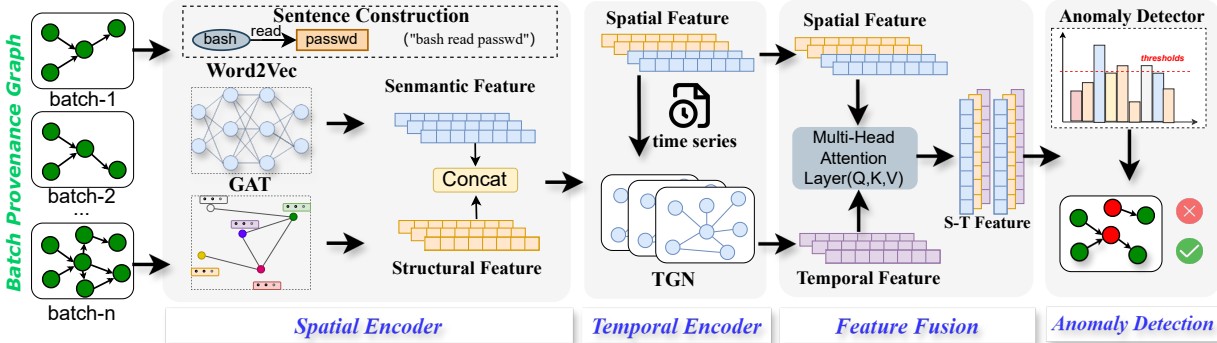

Figure 2: Overview of STGAN's architecture.

## 2.3 Threat Model

Our experiment environment relies on a Trusted Computing Base (TCB) comprising an operating system, an auditing framework, and provenance analysis tools. All components in the TCB are assumed to be fully functional from installation to completion, which is standard among existing provenance-based detectors. Hardware trojans and side-channel attacks that cannot be captured by audit are not considered in this paper. Additionally, the integrity of output audit data is assumed to be ensured by existing secure provenance and integrity audit systems [8, 35, 36, 50].

## 3 Design of STGAN

The overall workflow of STGAN is illustrated in Figure 2. STGAN receives streaming audit log input and segments the log information into multiple parts. For each subgraph within a segment, STGAN models and extracts both spatial and temporal information. In terms of spatial information extraction, STGAN encodes both semantic and structural features separately. First, it constructs sentences based on each node's first-hop neighbors and uses Word2Vec to learn the semantic features of the nodes. Then, GAT is applied to extract structural features, thereby forming a comprehensive spatial feature representation. For temporal information extraction, STGAN uses TGN to model temporal information and capture the temporal features of the nodes. Finally, STGAN employs a multi-head self-attention mechanism to fuse spatial and temporal features, generating a complete spatial-temporal graph embedding to support anomaly detection tasks. During the anomaly detection phase, STGAN utilizes XGBoost as the anomaly detector to perform anomaly detection.

## 3.1 Batch Provenance Graph Construction

STGAN first needs to convert system audit logs into a provenance graph. This graph represents the internal process interactions (such as *bash ->shell*), file operations (e.g., *bash -> /etc/passwd*), and network connections (e.g., *firefox ->101.162.12.201:8080*) within the host system. STGAN processes logs from Windows ETW or Linux Audit, which record process executions, file operations, and network connections on the host. To facilitate both temporal and spatial feature processing and support streaming detection, we designed a batch processing mechanism. Specifically, STGAN processes logs in

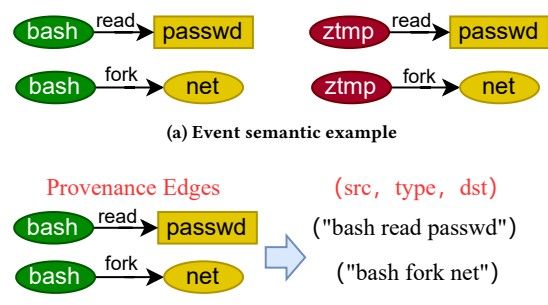

(a) Event semantic example

(b) Node sentence construction

Figure 3: Event semantic examples (a) and semantic information extraction (b)

configurable batches, where the batch size $k$ represents the number of audit logs processed per batch, and then converts each batch into a provenance graph.

## 3.2 Spatial Encoder

Nodes in a provenance graph contain rich attribute information that must first be mapped into a vector space for utilization by the model. Previous encoding schemes for provenance graphs have employed one-hot encoding for node and edge types or have utilized methods such as GAT and Graph Convolutional Networks (GCN)[24, 44], which are based on the assumption of homogeneity and primarily focus on structural features. However, these approaches have not fully leveraged the rich semantic information inherent in the graph. Unlike homogeneous graphs, such as social networks, the nodes and edges in a provenance graph possess specific naming information that often encapsulates significant behavioral semantics. For instance, as illustrated in Figure 3.(a), there is a notable distinction between a bash process (a user process) reading the passwd file and a ztmp process (a malicious process) accessing the same file. Although both instances represent a process reading a file within the provenance graph, their semantics are markedly different. Therefore, it is essential to consider this semantic information.

*3.2.1 Semantic Encoder.* We use a word2vec model for capturing semantic features of nodes. The training objective of Word2Vec can be expressed as maximizing the following log-likelihood function:

$$L = \sum_{w_t \in V} \sum_{-c \leq j \leq c, j \neq 0} \log P(w_{t+j} \mid w_t), \qquad (1)$$

where $V$ is the vocabulary, $w_t$ is the target word, $w_{t+j}$ is the context word, $c$ is the context window size, and $P(w_{t+j} \mid w_t)$ is the probability of predicting the context word $w_{t+j}$ given the target word $w_t$. This probability is typically computed using the softmax function:

$$P(w_{t+j} \mid w_t) = \frac{\exp(v_{w_{t+j}}^T \cdot v_{w_t})}{\sum_{w \in V} \exp(v_w^T \cdot v_{w_t})}, \qquad (2)$$

where $v_w$ is the vector representation of word $w$.

*3.2.2 Structural Encoder.* After extracting the semantic features, we further capture the structural features of nodes using Graph Attention Network (GAT). Unlike traditional graph convolutional networks, GAT introduces an attention mechanism that allows the model to aggregate information based on the importance of different neighbors. Initially, the attributes of nodes are one-hot encoded and mapped into a high-dimensional sparse vector space. Subsequently, for each node, GAT performs weighted aggregation of its feature vector through linear transformations and the attention mechanism to generate new node representations.

In STGAN, for each node $v_i$, we perform a linear transformation $W$ on its feature vector $h_i$ to obtain the linear representation $h_i' = Wh_i$. Then, GAT computes the attention coefficient $\alpha_{ij}$ between node $v_i$ and its neighbor $v_j$, which reflects the importance of neighbor node $v_j$ to node $v_i$. The attention coefficient $\alpha_{ij}$ is computed considering the similarity and relevance of node features, as follows:

$$\alpha_{ij} = \frac{\exp(\text{LeakyReLU}(a^T [Wh_i \| Wh_j]))}{\sum_{k \in \mathcal{N}(i)} \exp(\text{LeakyReLU}(a^T [Wh_i \| Wh_k]))}, \qquad (3)$$

where $a$ is a trainable attention weight vector, $\mathcal{N}(i)$ represents the neighbor set of node $v_i$, $\|$ denotes the vector concatenation operation, and LeakyReLU is used as a nonlinear activation function.

Finally, GAT aggregates the feature vectors of nodes and their neighbors using these attention coefficients to generate new node feature representations:

$$h_i'' = \sigma \left( \sum_{j \in \mathcal{N}(i)} \alpha_{ij} h_j' \right), \qquad (4)$$

where $\sigma$ is a nonlinear activation function.

By combining the semantic information extracted from Word2Vec with the structural information captured by GAT, we form a comprehensive spatial feature representation. This representation considers both the semantic features of the nodes and their positional relationships within the graph, providing a more thorough foundation for subsequent temporal encoding and anomaly detection.

## 3.3 Temporal Encoder

Due to the inherently dynamic nature of network attacks, it is essential to model the temporal information within provenance graphs to effectively capture these evolving activities. To achieve this, we employ the Temporal Graph Network (TGN) for temporal encoding. TGN is a network model specifically designed for dynamic graphs, allowing it to account for the influence of temporal information on the relationships between nodes and edges in graph-structured data. By using TGN, we can obtain temporal representations of each node at various time points, thereby enhancing our ability to track behavioral evolution and changes in anomaly patterns over time.

Specifically, TGN processes the spatial features of each batch graph organized in a time series. With its time-aware mechanism, TGN monitors interaction histories between nodes and aggregates this historical information to create time-aware node embeddings. These embeddings not only capture dynamic interactions among nodes but also reveal temporal behavior patterns within the system.

For each batch graph, a dynamic graph $G(t)$ is given, where $G(t)$ represents the graph at time $t$ and consists of a node set $\mathcal{V}$ and an edge set $\mathcal{E}$. Each edge $e \in \mathcal{E}$ has a timestamp $t_e$ indicating when it occurred. TGN performs temporal encoding through the following steps:

First, in the time embedding generation stage, the timestamp $t_e$ of each edge is embedded into a vector $\mathbf{t}_e$. This is done using positional encoding methods, where timestamps are mapped into a high-dimensional space using sine and cosine functions. Specifically, for a timestamp $t_e$, the time embedding $\mathbf{t}_e$ is calculated as:

$$\mathbf{t}_e = \left[ \sin\left( \frac{t_e}{10000^{2i/d}} \right), \cos\left( \frac{t_e}{10000^{2i/d}} \right) \right]_{i=1}^{d/2}, \qquad (5)$$

where $d$ is the dimension of the embedding space and $i$ is the positional encoding dimension index.

Next, in the time-aware information aggregation process, the time-aware feature $\mathbf{h}_i(t)$ of each node $v_i$ is updated by aggregating the information from its neighboring nodes. Specifically, the feature representation $\mathbf{h}_i(t)$ of node $v_i$ at time $t$ is updated by aggregating the features of neighboring nodes and the time embeddings of the edges connecting them. For a neighbor node $v_j$ and an edge $e_{ij}$, the update formula is:

$$\mathbf{h}_i(t) = \text{AGGREGATE}\left( \left\{ \mathbf{h}_j(t) \| \mathbf{t}_{e_{ij}} \mid v_j \in \mathcal{N}(v_i) \right\} \right), \qquad (6)$$

where AGGREGATE is the aggregation function, $\|$ denotes the feature concatenation operation, and $\mathcal{N}(v_i)$ represents the neighbor set of node $v_i$.

Finally, in the node embedding update stage, the embedding $\mathbf{h}_i(t)$ of each node at the end of time step $t$ is updated to a new time-aware embedding. The update function typically involves a fully connected layer and an activation function to fuse the state from the previous time step with the current time step features. The update formula is:

$$\mathbf{h}_i(t + 1) = \text{ReLU}\left( W \cdot \mathbf{h}_i(t) + b \right), \qquad (7)$$

By these steps, TGN generates time-aware node embeddings $\mathbf{h}_i(t)$ that reflect the dynamic behavior and temporal evolution of nodes in the graph. These time-aware embeddings are further enhanced by multi-head self-attention mechanisms to better capture long-term temporal dependencies.

## 3.4 Multi-dimensional Feature Fusion

To effectively integrate the spatial features derived from both semantic vectors (via Word2Vec) and structural features (via GAT), along with the temporal features extracted using TGN, we apply a multi-head self-attention mechanism. The motivation behind this approach is to ensure that the combined spatial-temporal features can fully capture the temporal relationships between nodes while preserving the rich information in both the spatial and temporal domains. By utilizing multi-head self-attention, we enable the model to focus on different aspects of the spatial-temporal features from various perspectives, leading to more accurate and robust representations for anomaly detection.

Formally, given the spatial vector $X_s$ (comprising both Word2Vec and GAT features) and the temporal vector $X_t$ derived from TGN, we compute the query, key, and value matrices for each attention head $h$:

$$Q_h = W_{Q_h}[X_s \| X_t], \quad K_h = W_{K_h}[X_s \| X_t], \quad V_h = W_{V_h}[X_s \| X_t], \tag{8}$$

where $W_{Q_h}$, $W_{K_h}$, and $W_{V_h}$ are the learned projection matrices for head $h$, and $\|$ denotes concatenation.

For each head, the attention scores are computed using the scaled dot-product attention mechanism:

$$\text{Attention}_h(Q_h, K_h, V_h) = \text{softmax}\left(\frac{Q_h K_h^\top}{\sqrt{d_k}}\right) V_h, \tag{9}$$

where $d_k$ is the dimension of the key vectors. The attention scores allow the model to assign different weights to various temporal and spatial features, dynamically focusing on the most relevant aspects.

After calculating the attention output for each head, we concatenate the outputs from all heads and project them to form the final representation:

$$Z = W_O \left[ \big\|_{h=1}^{H} \text{Attention}_h(Q_h, K_h, V_h) \right], \tag{10}$$

where $W_O$ is the learned output projection matrix, and $H$ represents the number of attention heads.

The final output $Z$ represents the fused spatial-temporal features, capturing multiple relationships across temporal and spatial dimensions. This multi-head mechanism enhances the model's ability to learn spatial-temporal feature patterns, improving the performance in downstream anomaly detection tasks. Specifically, this mechanism helps in detecting anomalies by capturing long-term dependencies across time and space, which is crucial for detecting slow, evolving cyberattacks that might span across multiple time steps.

## 3.5 Anomaly Detection

In our approach, anomaly detection is a critical component for identifying potential anomalies and malicious activities within the system. To achieve this, we have selected the lightweight detector XGBoost [10] as the core module for anomaly detection.

XGBoost iteratively trains decision trees, where each tree corrects the errors of the previous ones to reduce overall prediction error. In our task, the input features for XGBoost include the fused spatial and temporal features extracted from Word2Vec, GAT, and TGN. Instead of directly predicting if a node is malicious or normal, XGBoost is used to predict the node type. By predicting node types, the system can identify behavior deviations from typical patterns, which can indicate potential anomalies.

The trained XGBoost model leverages these multi-dimensional features to enhance the accuracy and robustness of anomaly detection. By focusing on predicting node types, the model is able to identify subtle deviations in node behavior that could indicate potential anomalies, offering a more precise method for detecting malicious activities within the system.

## 4 Evaluation

### 4.1 Experiments Setup

**Dataset.** The DARPA Transparent Computing (TC) dataset is a high-fidelity cybersecurity dataset designed to simulate complex enterprise environments and advanced persistent threats (APTs). This dataset focuses on multi-stage attacks within a realistic network architecture, capturing detailed host and network events, including both normal and malicious activities. It is ideal for evaluating anomaly detection methods and analyzing real-world attack behaviors, providing rich metadata such as timestamps, event types, and related naming information for comprehensive threat analysis. Considering our need for semantic information, we selected the more comprehensive Trace, Theia, and Cadets subsets. The detail of dataset is illustrated in table 2.

For more details on the experimental setup and baselines setup, please refer to Appendix A.

### 4.2 Comparison Experiments

In this section, we evaluate the performance of STGAN across three datasets and compare it with previous detectors. Table 1 presents our experimental results, where STGAN consistently performs the best in the Trace, Cadets, and Theia scenarios, with F1 scores exceeding 97%. In contrast, the F1 score of the static feature-based detector HOLMES is only 2.45%, the spatial feature-based detector FLASH achieves a maximum of 95%, and the temporal feature-based detector TGN reaches 95% as well. We explain the reasons behind these results as follows: The early HOLMES detector, based on the Apt lifecyle model [4], constructs a set of TTP rules to match patterns in the graph, which results in high recall but low precision (less than 5%) because these rules are often triggered by normal behaviors, leading to a high false positive rate. Among the spatial feature-based detectors, Threatrace performed the worst because it relies on GraphSAGE to capture spatial features but neglects semantic and spatial-temporal features, which weakens its detection performance. FLASH performs better as it first captures semantic features through Word2Vec and then uses these as initial embeddings fed into GCN to capture structural features. Among the spatial-temporal feature-based detectors, Unicorn and AnoGraph detect anomalies by modeling frequency information in temporal graphs, but these features are inherently low-dimensional and lack rich spatial information, resulting in poor outcomes (with an average F1 score below 50%). Our implementation of the traditional TGN model shows better results than Unicorn and AnoGraph due

**Table 1: Comparison Experiments**

| Model | Theia | | | | Cadets | | | | Trace | | | |
|---|---|---|---|---|---|---|---|---|---|---|---|---|
| | Precision | Accuracy | Recall | F1 | Precision | Accuracy | Recall | F1 | Precision | Accuracy | Recall | F1 |
| HOLMES [32] | 0.0120 | 0.9961 | 0.9900 | 0.0230 | 0.0126 | 0.9557 | 1.0000 | 0.0245 | 0.0120 | 0.9825 | 1.0000 | 0.0220 |
| Unicorn [18] | 0.6700 | 0.8000 | 0.6700 | 0.6700 | 0.3100 | 0.4400 | 1.0000 | 0.4700 | 0.2800 | 0.4300 | 1.0000 | 0.3400 |
| AnoGraph [9] | 0.2319 | 0.9066 | 0.4619 | 0.3088 | 0.0606 | 0.8895 | 0.4698 | 0.2133 | 0.0294 | 0.7597 | 0.4698 | 0.0553 |
| Threatrace [44] | 0.7156 | **0.9968** | 0.9999 | 0.8336 | 0.9035 | 0.9903 | 0.9997 | 0.9526 | 0.8701 | 0.9994 | 0.9963 | 0.9415 |
| FLASH [38] | 0.9203 | 0.9944 | 0.9987 | 0.9519 | 0.9412 | 0.9998 | 0.9999 | 0.9605 | 0.9501 | **0.9989** | 0.9999 | 0.9703 |
| TGN [39] | 0.9091 | 0.9967 | **1.0000** | 0.9524 | 0.8000 | 0.9940 | **1.0000** | 0.8889 | 0.8732 | 0.9911 | **1.0000** | 0.9284 |
| GCN [26] | 0.8294 | 0.9844 | 0.9983 | 0.9060 | 0.8959 | 0.9925 | 0.9984 | 0.9444 | 0.9090 | 0.9937 | 0.9840 | 0.9381 |
| STGAN(ours) | **0.9449** | 0.9956 | 0.9983 | **0.9709** | **0.9840** | **0.9994** | 0.9983 | **0.9911** | **0.9950** | 0.9991 | 0.9883 | **0.9916** |

**Table 2: DARPA E3 Dataset Description**

| Dataset | Edges | Nodes | Malicious | Size (GB) |
|---|---|---|---|---|
| Trace | 4,080,457 | 3,220,596 | 68,082 | 18.6 |
| Cadets | 3,303,264 | 1,614,189 | 12,846 | 16.8 |
| Theia | 10,929,710 | 3,505,326 | 25,362 | 31.6 |
| Total | 18,313,431 | 8,340,111 | 106,290 | 67 |

to its improved modeling of node spatial information (achieving an F1 score greater than 90%).

Overall, STGAN achieved the best performance, attributed to its pioneering use of spatial-temporal graphs to model provenance graphs, effectively integrating spatial and temporal features to produce higher-quality and more discriminative feature representations.

## 4.3   Ablation Study

In this section, we conduct an ablation study on STGAN to explore the impact of each component on the overall performance. Specifically, we removed or replaced certain components of STGAN and performed evaluations. Table 3 presents our experimental results, where X, Y, A, and B represent the major components of STGAN, namely: X: GAT encoder, Y: Word2Vec, A: TGN encoder, and B: Multi-Head Attention mechanism.

As the results show, the absence of any component leads to a decline in detection performance, highlighting the rationality of STGAN's design. For provenance graphs, both structural and semantic information contain rich characteristics of malicious and benign behaviors. This is reflected in the performance of groups (X, A, B) and (Y, A, B), where the lack of structural or semantic information leads to a decrease in detection accuracy. Spatial-temporal features capture the evolving characteristics of the provenance graph over time, and the results of group (X, Y) demonstrate that the spatial-temporal characteristics of malicious behavior differ from those of benign behavior. Benign behaviors tend to exhibit stable temporal patterns, while malicious behaviors often adopt irregular changes, such as launching multiple actions within a short period during scanning activities, which leads to significant differences in the temporal feature space. Feature fusion is equally important for STGAN. After removing the Multi-Head Attention mechanism, the results of group (X, Y, A) show a decrease in detection performance, indicating that feature fusion helps the model better weigh the importance

of multiple features, resulting in improved detection performances. Furthermore, we replaced some components to further evaluate STGAN, using GCN instead of GAT and Self-Attention instead of Multi-Head Attention. The results show a decrease in detection performance, which can be attributed to GAT's ability to capture more complex node relationships compared to GCN, and Multi-Head Attention's ability to handle multiple feature perspectives more effectively than Self-Attention.

**Feature visualisation analysis.** To illustrate STGAN's comprehensiveness and explainability in feature extraction, we performed a t-SNE visualization on node embeddings from the TRACE dataset. Figure 4 reveals that the full STGAN model (a) achieves the clearest separation of benign and malicious nodes, indicating that the combination of all components—GAT, Word2Vec, TGN, and Multi-Head Attention—effectively captures spatial, temporal, and semantic features.

When examining the partial models that exclude one component, shown in (b) through (e), we observe weaker clustering and less distinct separation, which underscores the importance of each module. Specifically, in (b), the exclusion of GAT hinders the model's ability to capture structural relationships, as nodes lose the context of their connectivity patterns, leading to dispersed clusters. In (c), the absence of Word2Vec affects the model's capacity to discern semantic nuances between nodes, resulting in reduced clarity in how nodes are grouped. Similarly, in (d), the removal of TGN limits the model's temporal encoding, which diminishes its ability to capture behavioral changes over time, making benign and malicious nodes more intermixed. Finally, in (e), without Multi-Head Attention, the model struggles to effectively integrate the spatial and temporal features, weakening its overall feature representation and making the boundaries between benign and malicious nodes less defined.

Overall, STGAN achieves the best results by effectively extracting and fusing the spatial-temporal and temporal features of the provenance graph for detection. Through the ablation study, we have also demonstrated the rationality of the key module designs in STGAN.

## 4.4   Parameter Sensitivity Analysis

In this section, we evaluate the impact of hyper-parameter settings on the performance of STGAN. We mainly focus on the batch processing size $k$, learning rate $lr$, and embedding dimension $d$. To ensure fairness, the TRACE dataset is primarily used for our experiments. Figure 5 provides the experimental results.

**Table 3: Ablation Experiments, X,Y,A,B denote GAT encoder, Word2Vec, TGN, and Multi-Head Attention, respectively.**

| Method | Theia | | | | Cadets | | | | Trace | | | |
|---|---|---|---|---|---|---|---|---|---|---|---|---|
| | Precision | Accuracy | Recall | F1 | Precision | Accuracy | Recall | F1 | Precision | Accuracy | Recall | F1 |
| X,Y,A | 0.6142 | 0.9512 | 0.9984 | 0.7605 | 0.8126 | 0.9904 | 0.9995 | 0.8964 | 0.9160 | 0.9942 | 0.9884 | 0.9509 |
| X,Y | 0.7830 | 0.9786 | 0.9983 | 0.8777 | 0.8152 | 0.9909 | 0.9995 | 0.8980 | 0.8450 | 0.9890 | 0.9884 | 0.9111 |
| Y,A,B | 0.8213 | 0.9833 | 0.9983 | 0.9012 | 0.8186 | 0.9916 | 0.9995 | 0.9000 | 0.6644 | 0.9710 | 0.9884 | 0.7949 |
| X,A,B | 0.3123 | 0.8249 | 0.9984 | 0.4758 | 0.4701 | 0.9446 | 0.9995 | 0.6394 | 0.5473 | 0.9523 | 0.9884 | 0.7045 |
| X,Y,A,B (Self-Attention) | 0.8294 | 0.9844 | 0.9983 | 0.9060 | 0.8959 | 0.9956 | 0.9984 | 0.9444 | 0.9032 | 0.9933 | 0.9884 | 0.9439 |
| X (GCN),Y,A,B | 0.3943 | 0.8789 | 0.9984 | 0.5654 | 0.8273 | 0.9922 | 0.9988 | 0.9050 | 0.6648 | 0.9710 | 0.9884 | 0.7949 |
| STGAN(ours) | **0.9449** | **0.9956** | 0.9983 | **0.9709** | **0.9840** | **0.9994** | 0.9983 | **0.9911** | **0.9950** | **0.9991** | 0.9883 | **0.9911** |

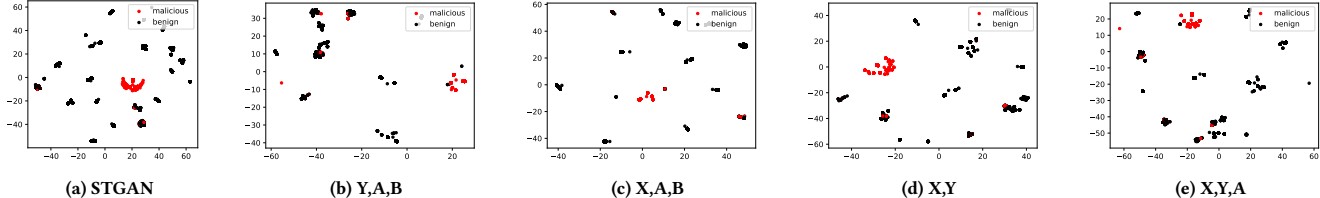

(a) STGAN     (b) Y,A,B     (c) X,A,B     (d) X,Y     (e) X,Y,A

**Figure 4: We demonstrate the feature extraction effectiveness under different components (X: GAT, Y: Word2Vec, A: TGN, B: Multi-Head Attention) through t-SNE visualization. It can be observed that STGAN achieves the best feature space differentiation.**

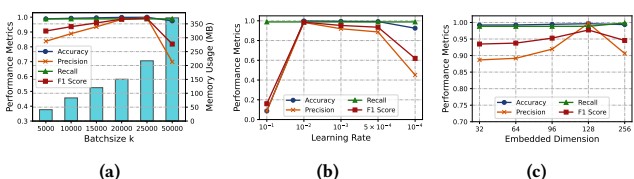

(a)     (b)     (c)

**Figure 5: (a),(b),(c) show the result of batch-size $k$, learn rate $lr$, and embedding dimension $d$, respectively.**

**Batch size $k$.** From these results, it is evident that batch-size significantly influences model performance. As batch-size increases, the model's Accuracy and Precision first increase and then decrease, while Recall remains relatively stable. The model performs relatively well with smaller batch-sizes (5000 to 10000), with Precision and F1score gradually improving as the batch-size grows. When the batch-size reaches 25000, the model shows the best balance, with Precision reaching 0.995 and F1score reaching 0.9916, indicating that the model's anomaly detection capability is strongest at this point. However, when the batch-size increases to 50000, both Precision and F1score drop significantly. This may be due to the large batch-size reducing the model's sensitivity to subtle differences. Therefore, selecting a moderate batch-size (such as 25000) during STGAN training can achieve optimal detection performance, while an overly large batch-size may lead to performance degradation.

**Learning rate $lr$.** As for the learning rate $lr$, when the learning rate decreases from 0.1 to 0.01 and 0.001, the model's Precision and F1 Score increase rapidly, reaching high performance levels. Notably, when the learning rate is 0.01, the model demonstrates optimal performance balance, with Precision reaching 0.9831 and F1

Score at 0.9857. However, when the learning rate is further reduced to 0.0005 and 0.0001, both Precision and F1 Score decline. This may be due to the fact that a too-low learning rate leads to very small update steps, resulting in slow convergence and the model being easily trapped in local optima, without effectively jumping out of these regions to approach the global optimum. Therefore, selecting a moderate learning rate (such as 0.01) during STGAN training can achieve optimal detection performance, while excessively high or low learning rates may lead to performance degradation.

**Embedding dimension $d$.** As for the embedding dimension $d$, we observe that as the embedding dimension increases, the performance metrics improve consistently, with the optimal performance achieved at 128 dimensions. Beyond this point, such as at 256 dimensions, further increases do not yield significant gains and may introduce diminishing returns. Therefore, an embedding dimension of 128 offers the best balance between performance and computational efficiency, capturing sufficient feature richness without unnecessary complexity.

## 5 Discussion and Limitations

**Multi-level feature extraction.** In provenance-based anomaly detection, static features such as structural features [24, 44], statistical features [21, 28], and prior knowledge [22, 32] have been widely used. Dynamic features have been mentioned more recently, but they either focus solely on dynamic statistical features [18] or consider only the dynamic changes in the graph [11, 47], neglecting spatial features. As attackers' behavior becomes increasingly complex, with long-term temporal variations, it is crucial to comprehensively model and extract information that captures both rich temporal and spatial characteristics. In this work, we introduce the concept of spatial-temporal graphs, extracting rich sequential, semantic, and structural information in both spatial and temporal

dimensions. Through feature fusion, we obtain high-quality embedding vectors, thereby achieving optimal detection performance. Specifically, by leveraging multi-head self-attention mechanisms, STGAN can capture complex temporal dependencies alongside spatial patterns, which allows it to effectively detect sophisticated multi-stage attacks. The attention mechanism ensures that the model can dynamically focus on the most relevant spatial-temporal features over time, providing effective detection even in evolving attack scenarios.

**Graph manipulation attack.** Graph manipulation attacks [16, 33, 40] have recently posed significant challenges for provenance-based anomaly detectors, as attackers may manipulate both the graph structure and its attributes, leading to evasion attacks. Therefore, it is essential to discuss the robustness of STGAN against such attacks. STGAN jointly models spatial and temporal features, ensuring that our detection vectors incorporate semantic information, structural information, and temporal variations. Evading STGAN would require mimicking and manipulating all three dimensions simultaneously, which is highly challenging for attackers. Even if attackers disrupt the structure by abusing system calls with malware, the semantic space can still flag anomalies, as it is difficult for attackers to imitate benign behavior in terms of semantics (e.g., system process names). The multi-head self-attention mechanism further strengthens STGAN's robustness, as it allows the model to focus on the most relevant interactions over both space and time, making it harder for attackers to fool the system across multiple dimensions. Hence, our multi-dimensional feature extraction and fusion confer robustness against graph manipulation attacks.

**Limitation.** Although STGAN is capable of fully extracting and fusing the features of provenance graphs across multiple dimensions, we must acknowledge the resource consumption of STGAN, despite employing batch graphs, lightweight anomaly detectors, and encoders. Efficiency and accuracy in host anomaly detection often involve a trade-off, where more precise detection typically comes with higher model complexity. Furthermore, we are exploring recent advancements in vector caching techniques [38], where pre-trained node embeddings can be stored in a cache for rapid reuse, reducing the need for retraining and accelerating inference times. While STGAN shows promise in terms of accuracy, practical deployment in resource-constrained environments would benefit from these ongoing optimizations.

## 6  Related Work

**Host Provenance Graph-based Anomaly Detection.** Host Provenance graphs have been widely applied in attack detection due to their rich semantic and contextual information [12, 23]. Recently, they have been categorized into three main types: dynamic graph learning approaches (temporal level), static graph learning approaches (spatial level), and approaches based on low-level static features (including statistical based and rule based methods). Statistical based methods [14, 21, 36] model the anomaly degree of nodes using features such as temporal correlation, degree distribution, and rarity. Rule-based methods [20, 22, 31, 32, 51] create rules based on external knowledge to incrementally match patterns in the provenance graph for anomaly detection. Static graph learning approaches [6, 19, 24, 41, 44, 49] include sequence learning methods

that extract and model sequence features for anomaly detection, as well as deep graph learning techniques like GAT [42] or GraphSAGE [17] to extract structural features for detection, along with node interaction features [49] for graph-level and edge-level detection. Dynamic graph learning approaches include sketch-based defenses and dynamic graph learning schemes. For example, Unicorn models the statistical characteristics of dynamic graphs, capturing temporal feature variations to detect anomalies, while ProGrapher [47] adopts Graph2Vec [34] to learn embeddings of streaming graphs and detects anomalies by comparing feature differences between consecutive graph snapshots. Overall, these approaches learn different feature representations of host provenance graphs from various dimensions to perform anomaly detection tasks. Compared to existing methods, STGAN maximally captures graph features across multiple dimensions, significantly enhancing the discriminative power of the learned embeddings in the feature space, resulting in more accurate and robust detection performance.

**Spatial Temporal Graph Neural Network.** Spatial-Temporal Graph Neural Networks (STGNNs) have gained significant traction in recent years for handling complex time-varying graph data across tasks such as traffic prediction, social network analysis, and recommendation systems [45]. STGODE [15] and PDFormer[25] have introduced advanced techniques to better capture spatial-temporal patterns. While early approaches like STGCN [48] relied on pre-defined graphs using domain-specific knowledge, recent work has shifted towards more flexible, self-learned methods. For example, Graph WaveNet [46] and AGCRN [7] dynamically generate adjacency matrices based on the underlying data, offering greater adaptability. Despite these advancements, most of these models are applied in domains such as transportation or communication networks, focusing on structural and temporal dependencies without considering deeper semantic layers.

Our approach differs from previous STGNN methods used in tasks like traffic prediction. We thoroughly analyzed the characteristics of provenance graphs and introduced semantic features using Word2Vec. By combining spatial and temporal, information, we comprehensively capture the interaction patterns of system entities within the provenance graph, enabling more effective detection of complex attack patterns that are often overlooked by traditional methods that rely solely on spatial-temporal features.

## 7  Conclusion

In this paper, we propose a novel method STGAN, which employs spatial-temporal graphs to model Host Provenance Graphs (HPGs) for enhanced anomaly detection. We observed that provenance graphs contain rich spatial and temporal information, which has not been fully leveraged in previous research. To address this, STGAN utilizes spatial and temporal encoders to comprehensively extract spatial-temporal features, including semantic, structural, and temporal characteristics. These features are then fused using a self-attention mechanism, resulting in a more robust feature representation. Our method demonstrated superior performance across four datasets, outperforming existing state-of-the-art methods in detecting advanced cyber threats.

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

## A More Experiments Details

### A.1 Implementation

During the graph data construction phase, we used a batch-based approach, setting the batch size to 20,000 log entries. We constructed a Word2Vec model [30] by forming sentences from the first-order neighbors of nodes and edge types. For structural feature extraction, we employed a two-layer GAT architecture. Overall, we used DGL [43] to build and train the graph model. For downstream detection tasks, we utilized scikit-learn [37] to build the classification model. The detailed training and detection process for STGAN is presented in Appendix B.

### A.2 Baseline Detector and Metrics.

We compared STGAN with several baseline methods, including the Graph Sketching-based Unicorn [18] and Anograph [9] approach, spatial feature-based methods such as Threatrace [44] and FLASH [38], and static low-dimensional feature-based HOLMES [32]. These methods represent approaches focusing on spatial features and those that incorporate temporal aspects. Additionally, we also implemented a GAT-based [42] and TGN-based [39] anomaly detection method for comparison. More descriptions of these baseline methods as follows.

- **HOLMES [32]**: This detector utilizes the APT lifecycle model to define a set of TTP rules for matching in the provenance graph, identifying nodes that conform to these rules as anomalous.
- **Unicorn [18]**: This detector employs sketching techniques, leveraging historical information to model and identify anomalous activities.
- **AnoGraph [9]**: Also based on sketching techniques, this detector defines a set of higher-order CMS structures and detects anomalous edges and subgraphs by identifying changes in local or global densities.
- **Threatrace [44]**: This detector uses GraphSAGE [17] to learn node embeddings and performs anomaly detection.

- **FLASH [38]**: This detector extracts semantic features of nodes using Word2Vec and then applies GAT for structural feature extraction, followed by anomaly detection.
- **GAT [42]**: This is a detector we implemented, which first learns node embeddings through GAT and then performs anomaly detection.
- **TGN [39]**: Another detector we implemented, it uses one-hot encoding for node types and learns node embeddings through TGN, followed by anomaly detection.

We evaluated the model using accuracy, precision, recall, and F1 score. Since DARPA did not provide well-labeled data for the TC dataset, but only raw log data and operational report files, we used the labeled data from previous studies [44] for model evaluation. Additionally, as the rule-based labeling system [32] does not have publicly available code, we implemented and tested the system based on the complete table provided in [32].

## B Pseudo-codes of STGAN

The overall framework of STGAN is available in Algorithm 1. This Pseudo-code implements STGAN that extracts spatial, temporal, and semantic features of graph nodes using GAT, TGN, and Word2Vec, respectively. It then combines these features via a multi-head self-attention mechanism to classify and detect malicious nodes in the graph.

---

**Algorithm 1** STGAN Feature Extraction and Detection

---

**Require:** Dataset $\mathcal{D}$, Word2Vec model $w2vmodel$, GAT model $G$, TGN model $T$, epochs $n$, node phrases $P$, labels $\mathbf{Y}$, graph edges $\mathcal{E}$, temporal edge attributes $\mathcal{T}$

**Ensure:** Detected malicious nodes $\mathcal{M}$

1: Initialize GAT $G$, TGN $T$, and Multi-Head Self-Attention model $\mathcal{A}$
2: Split dataset $\mathcal{D}$ into training set $\mathcal{D}_{train}$ and test set $\mathcal{D}_{test}$
3: **for** $epoch$ = 1 to $n$ **do**
4:   **for** each node $v \in \mathcal{D}_{train}$ **do**
5:     Extract phrase $p_v \in P$ for node $v$
6:     Compute Word2Vec embedding $\mathbf{w}_v = w2vmodel(p_v)$
7:     Apply positional encoding: $\mathbf{w}'_v = \text{PosEnc}(\mathbf{w}_v)$
8:     Concatenate GAT output $\mathbf{h}_v$ and $\mathbf{w}'_v$: $\mathbf{z}_v = [\mathbf{h}_v \parallel \mathbf{w}'_v]$
9:     Pass $\mathbf{z}_v$ through TGN: $\mathbf{z}'_v = T(\mathbf{z}_v, \mathcal{E}, \mathcal{T})$
10:    Fuse features via multi-head self-attention: $\mathbf{o}_v = \mathcal{A}(\mathbf{z}'_v)$
11:   **end for**
12: **end for**
13: Evaluate the model on $\mathcal{D}_{test}$
14: **for** each node $v \in \mathcal{D}_{test}$ **do**
15:   Extract phrase $p_v \in P$ and compute embeddings as in training
16:   Classify node $v$ and rank predictions
17: **end for**
18: **return** Detected malicious nodes $\mathcal{M}$

---

## C Time Complexity of STGAN

The time complexity of STGAN can be assessed by analyzing its key operations for feature extraction and detection on each node within the provenance graph. First, the complexity for computing

Word2Vec embeddings is $O(|V| \times d)$, where $|V|$ represents the vocabulary size and $d$ the embedding dimension, which are constant post-training. In the GAT, each node interacts with $k$ neighbors, leading to a complexity of $O(k \times d \times d')$; for the entire graph, this scales to $O(|N| \times k \times d \times d')$, where $|N|$ is the total number of nodes. Temporal features are extracted via the TGN with a complexity of $O(m \times t \times d'')$, where $m$ is the edge count, $t$ the number of time steps, and $d''$ the transformed feature dimension.

The multi-head self-attention mechanism for feature fusion has a complexity of $O(h \times d \times |N|)$, where $h$ is the number of attention heads. Therefore, the overall time complexity per epoch for STGAN is: $O(n \times (|N| \times k \times d \times d' + m \times t \times d'' + h \times d \times |N|))$ This structure allows STGAN to scale effectively for streaming provenance graph environments, with adjustable batch sizes that help manage computational costs efficiently.

