# OpenReview forum: "STGAN: Detecting Host Threats via Fusion of Spatial-Temporal Features in Host Provenance Graphs"
_ACM.org/TheWebConf/2025/Conference — WWW 2025 Oral_

### Official Review · Reviewer_c5kg · 2024-11-29

**Novelty:** 5
**Technical Quality:** 5

**Review:**

This paper presents STGAN, an approach for detecting host threats by fusing spatial-temporal features in host provenance graphs. The proposed approach is quite well-presented, with comprehensive experiments. While there are some limitations in terms of scalability analysis and security guarantees, these could be addressed in future work. Upon revision this work could be accepted for publication if the below recommendations are satisfied. My recommendations are the followings:
1. Expand scalability testing
2. Add more implementation details if possible include real-world case studies
3. Expand limitations discussion

**Questions:**

1. What are the practical scaling limitations?
2. What is the real-time detection latency in practice?
3. How does the batch size affect detection time vs accuracy trade-offs?
4. Can you provide more details on the training data requirements and model update strategies for deployment?

**Reviewer Confidence:**

4: The reviewer is certain that the evaluation is correct and very familiar with the relevant literature

**Scope:**

3: The work is somewhat relevant to the Web and to the track, and is of narrow interest to a sub-community

---

### Official Review · Reviewer_hgbA · 2024-11-30

**Novelty:** 5
**Technical Quality:** 5

**Review:**

Strengths:
1. The paper proposes a host-based threat detection method that integrates spatiotemporal features, which effectively improves the overall performance of host-based threat detection, especially in the aspect of spatiotemporal feature modeling, showcasing strong innovation.
2. Experimental results on multiple benchmark datasets demonstrate that the proposed method significantly outperforms existing approaches, indicating its potential for real-world applications.
3. The paper provides a thorough modular analysis of the model, along with sensitivity testing for its parameters, validating the effectiveness of each component and offering reasonable optimization suggestions for further improvement.
Weaknesses:
1. The model was trained using a relatively small number of benchmark datasets, only three in total, which may lead to overfitting and negatively affect the model's generalization capability on unseen data. To enhance the robustness of the model, it is recommended to incorporate additional diverse datasets for training and evaluation.
2. There is a lack of assessment regarding the model's practical applicability, particularly in terms of its parameter size, FLOPs (floating point operations), and computational time costs. These factors are crucial for understanding the model's real-world performance and should be addressed in future work.

**Questions:**

When the time span of the log data is very long, the number of nodes and edges can grow explosively, which places significant computational strain on both the training and inference processes of the Temporal Graph Network (TGN) model. Addressing this challenge, such as by reducing computational complexity while maintaining model performance, is a critical issue in the current research. It is recommended to explore data reduction techniques, sampling strategies, or optimizations in graph neural networks to mitigate this problem.

**Reviewer Confidence:**

1: The reviewer's evaluation is an educated guess

**Scope:**

3: The work is somewhat relevant to the Web and to the track, and is of narrow interest to a sub-community

---

### Official Review · Reviewer_AAfk · 2024-12-01

**Novelty:** 5
**Technical Quality:** 5

**Review:**

As the title of the paper indicates, the authors proposed to integrate spatial-temporal information into host provenance graph modeling such that more accurate detection of host threats can be achieved. Details of the implementation methods are presented and some comparisons versus existing methods have been made. While it is intuitive that integrating spatial-temporal information shall help improve detection accuracy, authors' discussions on implementation methods are detailed and interesting. The paper is well written, with few typos (like the "three three" on page 2, which can be easily identified and corrected).

As correctly pointed out by the authors, there is a tradeoff between computational complexity and algorithm performance. The discussions on the comparisons between the proposed method and the existing methods however have been mainly on performance, while it is not clear how much the additional overhead the proposed method may impose versus some state-of-the-art existing methods.

**Questions:**

1. From Table 1, it appears that FLASH is a good competitors against the proposed method STGAN. Could the additional overhead of the proposed method (if any) be well justified by the marginal (?) improvement in performance? In other words, could we justify that the proposed method has achieved better trade-off compared with some existing methods, please?

**Reviewer Confidence:**

3: The reviewer is confident but not certain that the evaluation is correct

**Scope:**

3: The work is somewhat relevant to the Web and to the track, and is of narrow interest to a sub-community

---

### Official Review · Reviewer_xRyi · 2024-12-02

**Novelty:** 3
**Technical Quality:** 3

**Review:**

### **Recommendation**: Reject

---

This paper introduces **STGAN**, a spatio-temporal graph-based anomaly detection model designed to detect advanced cyberattacks, such as Advanced Persistent Threats (APTs) and ransomware, using Host Provenance Graphs (HPGs). The proposed framework aims to overcome the limitations of single-dimensional feature analysis by integrating spatial, temporal, and semantic features. Although the model demonstrates promising detection performance on DARPA TC datasets, the paper has critical shortcomings in terms of novelty, generalizability, and experimental rigor.

---

### **Strengths**

1. **Relevant Problem Domain**:
    - The paper addresses a pressing issue of increasing cyberattacks and the limitations of existing detection approaches. This aligns well with current challenges in cybersecurity.
2. **Well-Structured Presentation**:
    - The paper systematically describes the STGAN framework, making it accessible to readers unfamiliar with the specific methods. Key components such as Word2Vec, GAT, and TGN are clearly introduced for semantic, spatial, and temporal feature extraction.
3. **Performance on Benchmark Datasets**:
    - STGAN achieves strong results on DARPA TC datasets, with F1-scores exceeding 97%, surpassing SOTA methods such as Threatrace and AnoGraph.
4. **Reproducibility and Open Resources**:
    - The authors contribute datasets and code to the community, enabling reproducibility and supporting further research in graph-based anomaly detection.

---

### **Weaknesses**

### **1. Lack of Significant Technical Contributions**

- STGAN largely combines existing methods without introducing substantial innovations. Each component is implemented in a standard way, with insufficient detail or justification for the design choices:
    - **Semantic Encoder**:
        - Word2Vec is used, but the authors fail to specify which training method (Continuous Bag of Words or Skip-gram) was employed. Furthermore, there is no rationale provided for this choice or its suitability for capturing semantic features in Host Provenance Graphs.
    - **Temporal Encoder**:
        - Sinusoidal encoding is adopted, but the paper does not explain why this specific encoding is appropriate for temporal feature extraction in the given context.
    - **Anomaly Detection Stage**:
        - XGBoost is employed as the final detection method, but its advantages over other potential classifiers (e.g., Random Forest or Neural Networks) are not discussed.
    - These omissions raise concerns about the depth of the technical contribution and the robustness of the design.

### **2. Generalizability and Scalability**

- The evaluation scope is limited to the DARPA TC datasets, which do not fully represent real-world scenarios or diverse graph structures.
    - The absence of experiments on larger graphs, streaming data, or datasets from other domains raises concerns about the generalizability of STGAN.
- **Parameter Sensitivity Analysis**:
    - While the paper includes a sensitivity analysis of key hyperparameters such as batch size, learning rate, and embedding dimensions, the results appear to reflect the specific characteristics of the TRACE dataset rather than generalizable patterns.
    - For example:
        - The **optimal batch size (25,000)**, learning rate (0.01), and embedding dimension (128) were determined based solely on TRACE data, which may not hold true for other datasets or graph structures.
        - Performance degradation with larger batch sizes or extreme learning rates suggests potential overfitting to specific dataset characteristics, undermining robustness.
    - Without testing on additional datasets, it is unclear whether these hyperparameter settings would transfer effectively to new, unseen scenarios.
- The scalability of STGAN to handle larger graphs or real-time data streams is not explored, leaving its practical applicability in resource-constrained environments uncertain.

### **3. Insufficient Discussion of Practical Applicability**

- The computational complexity introduced by multiple encoders (Word2Vec, GAT, TGN) and the self-attention mechanism is not analyzed.
    - The paper provides no information on detection latency or resource usage, making it unclear whether STGAN is feasible for real-time deployment.
    - Without overhead analysis, it is difficult to determine how well the method scales in resource-constrained environments.

### **4. Inconsistencies in Writing and Methodology**

- The manuscript contains inaccuracies and contradictions that reduce its clarity:
    - For instance, the introduction claims Word2Vec is used for structural features and GAT for semantic features, while the design section states the opposite.
    - Such inconsistencies cast doubt on the authors’ understanding and explanation of their methodology.
- The choice of specific configurations (e.g., Word2Vec training method, sinusoidal encoding) is not justified, leaving key design decisions unexplained.

---
### **Overall Assessment**

While STGAN addresses an important problem in cyberattack detection and achieves strong results on benchmark datasets, the paper lacks sufficient novelty, fails to demonstrate generalizability, and provides limited evidence of robustness and practical applicability. Key weaknesses include:

- The absence of novel technical contributions beyond the combination of existing methods.
- A narrow evaluation scope that does not consider real-world scenarios or diverse datasets.
- Missing discussions on computational efficiency, detection latency, and scalability.
- A lack of robustness validation against adversarial attacks or graph manipulation.

Given these significant shortcomings, I recommend **Reject**. The paper requires substantial revisions, particularly in terms of experimental scope, practical evaluation, and the introduction of meaningful innovations, to meet the standard for publication.

**Questions:**

### **Questions for the Authors**
1. **Technical Design Choices**:
    - Why was Word2Vec chosen for semantic encoding, and which training method (CBOW or Skip-gram) was employed? What evidence supports its suitability for the given task?
    - Why was sinusoidal encoding adopted in the Temporal Encoder? Have alternative temporal encoding techniques been considered or tested?
    - What is the rationale behind using XGBoost for anomaly detection? How does it compare to other classifiers like Random Forest or Neural Networks?
2. **Generalization Beyond DARPA TC**:
    - Have you tested STGAN on larger graphs, streaming data, or datasets outside of DARPA TC? How does the model perform in domains with significantly different graph structures or real-world variability?
3. **Computational Efficiency**:
    - What is the computational overhead of STGAN compared to baseline methods? Specifically:
        - How much detection latency does the model introduce?
        - How does STGAN’s resource usage scale with graph size and complexity?
4. **Robustness Against Attacks**:
    - Have you evaluated the robustness of STGAN against adversarial or graph manipulation attacks? Can you provide experimental evidence for the claim that attackers cannot mimic all three dimensions simultaneously?
5. **Parameter Sensitivity**:
    - How do key hyperparameters, such as the number of attention heads(H) or tree-related parameters in XGBoost, affect the performance of STGAN?

**Reviewer Confidence:**

3: The reviewer is confident but not certain that the evaluation is correct

**Scope:**

4: The work is relevant to the Web and to the track, and is of broad interest to the community